# A priori guarantees of finite-time convergence for Deep Neural Networks

## Abstract

In this paper, we perform Lyapunov based analysis of the loss function to derive an a priori upper bound on the settling time of deep neural networks. While previous studies have attempted to understand deep learning using control theory framework, there is limited work on a priori finite time convergence analysis. Drawing from the advances in analysis of finite-time control of non-linear systems, we provide a priori guarantees of finite-time convergence in a deterministic control theoretic setting. We formulate the supervised learning framework as a control problem where weights of the network are control inputs and learning translates into a tracking problem. An analytical formula for finite-time upper bound on settling time is provided a priori under the assumptions of boundedness of input. Finally, we prove that our loss function is robust against input perturbations.

## 1 Introduction

Over the past decade, Deep neural networks have achieved human-like performance in various machine learning tasks, such as classification, natural language processing and speech recognition. Despite the popularity of deep learning, the underlying theoretical understanding remains relatively less explored. While attempts have been made to develop deep learning theory by drawing inspiration from other related fields such as statistical learning and information theory, a comprehensive theoretical framework is still in an early developmental stage. It is difficult to perform mathematical analysis on Deep neural networks due to the large number of parameters involved. Other problems in deep neural networks revolve around the stability and desired convergence rate of the training. Since the performance of the network depends highly on the training data and the choice of the optimization algorithm, there is no guarantee that the training will converge. Our work attempts to give finite-time convergence guarantees for training of a deep neural network by utilizing an established stabilization framework from control theory.

Existing works in deep learning theory have attempted to bridge the gap in understanding deep learning dynamics by focusing on simple models of neural networks [Saxe et al. (2013), Li & Yuan (2017), Arora et al. (2018), Jacot et al. (2018)]. This could be attributed to the fact that current state-of-the-art deep learning models are highly complex structures to analyze. Jacot et al. (2018) proved that a multilayer fully-connected network with infinite width converges to a deterministic limit at initialization and the rate of change of weights goes to zero. Saxe et al. (2013) analyzed deep linear networks and proved that these networks, surprisingly, have a rich non-linear structure. The study shows that given the right initial conditions, deep linear networks are a finite amount slower than shallow networks. Following this work, Arora et al. (2018) proved the convergence of gradient descent to global minima for networks with dimensions of every layer being full rank in dimensions. While these studies give important insights into the design of neural network architecture and the behavior of training, their results may need to be modified in order to provide convergence guarantees for the conventional deep neural networks. Du et al. (2018) extended the work of Jacot et al. (2018) further by proving convergence for gradient descent to achieve zero training loss in deep neural networks with residual connections.

When it comes to convergence of certain state variables of a dynamical system, control theory provides a rich mathematical framework which can be utilized for analyzing the non-linear dynamics of deep learning [Liu & Theodorou (2019)]. One of the early works relating deep learning to control theory was of LeCun et al. (1988), which used the concept of optimal control and formulated

back-propagation as an optimization problem with non-linear constraints. Non-linear control has gained increasing attention over the past few years in the context of neural networks, especially for recurrent neural networks [Allen-Zhu et al. (2019), Xiao (2017)] and reinforcement learning [Xu et al. (2013), Gupta et al. (2019), Wang et al. (2019), Kaledin et al. (2020)]. A new class of recurrent neural networks, called Zhang Neural Networks (ZNN), was developed that expressed dynamics of the network as a set of ordinary differential equations and used non-linear control to prove global or exponential stability for time-varying Sylvester equation [Zhang et al. (2002), Guo et al. (2011)]. Li et al. (2013) introduced the sign bi-power activation function for Zhang Neural Networks (ZNN) which helps to prove the existence of finite-time convergence property. Haber & Ruthotto (2017) presents deep learning as a parameter estimation problem of non-linear dynamical systems to tackle the exploding and vanishing gradients.

The focus of this paper is on deriving a priori guarantee of attaining finite-time convergence of training in a supervised learning framework under some assumptions on inputs. The novelty lies in the fact that the weight update is cast as a finite-time control synthesis such that the loss function is proven to be a valid Lyapunov function. The resulting training update is derived as a function of time such that it ensures the convergence of Lyapunov function in finite time. The only assumption used is that the magnitude of atleast one input is greater than zero. Thus, the learning problem is converted into a finite time stabilization problem as studied rigorously in Bhat & Bernstein (2000). The contributions of the proposed study are twofold. First, we propose a Lyapunov candidate function to be used as loss function. Second, we modify the weight update of the neural network in such a way that the supervised training is converted into a dynamical control system. This allows us to use results from Bhat & Bernstein (2000) to a priori guarantee finite time convergence on the training. To the best of our knowledge, a guarantee of finite-time convergence is being studied for the first time in context of training a general multi-layer neural network. The proposed results will enable time bound training that will be useful in real-time applications.

The paper is organized as follows. Section 2 starts with introducing the Lyapunov function from the control theory perspective. Section 2.1 derives the weight update and lyapunov loss function for a single neuron case and proves that it satisfies the conditions required for finite-time stability theorems developed in Bhat & Bernstein (2000) to be applicable. Section 2.2 then proves that a similar result extends to a multi-layer perceptron network under reasonable assumptions on the input. In Section 2.3, we state the equations to compute upper bounds on the convergence time for training neural networks. Section 2.4 provides an extension to the case when bounded perturbations are admitted at the input and convergence guarantees are shown to hold true. In Section 3, some numerical simulations are presented for both single neuron and multi-layer perceptron cases for regression. Section 4 collects conclusions and discusses future scope.

## 2 PROPOSED METHOD TO CONVERT SUPERVISED LEARNING INTO DYNAMICAL CONTROL SYSTEM

This section motivates the development of a priori bounds on settling time with certain assumptions on the input. The weight update problem for supervised learning in neural networks is similar to the tracking problem of non-linear control systems. The idea is to synthesize a feedback control law based on certain Lyapunov function $V(x)$. A Lyapunov function is a positive definite function, i.e. $V(x) > 0$, and its time derivative is negative definite along the given system dynamics, i.e. $\dot{V} < 0$. We cast the supervised learning problem as a dynamical system which admits a valid Lyapunov function as the loss function with the weight update is designed as a function of time.

### 2.1 SINGLE NEURON CASE

We start with a simplistic single neuron case. Let $x \in \mathbb{R}^n$ be the input to the network where $x = \begin{bmatrix} x_1 & x_2 & \cdots & x_n \end{bmatrix}^\top$, $|x_i| < c, i = 1, 2, \cdots n$ holds true for some a priori but arbitrary scalar $c \in (0, \infty)$. Let $y^\star$ be the target output and $z = \sum_{i=1}^n w_i x_i + b$ be the linear combination of weights $w_i$ with inputs $x_i$ and bias $b$. The definition of signum function used here is $\text{sign}(x) = 1, \forall x > 0, \text{sign}(x) = -1, \forall x < 0, \text{sign}(x) \in [-1, 1], x = 0$. For our analysis, we choose sigmoid function as our activation function, i.e. $\sigma(z) = \frac{1}{1+e^{-z}}$. The output of the neural network is given by $y = \sigma(z)$. Let the error in output be defined as $\bar{e} = y - y^*$. The first objective is to convert our

loss function into a candidate Lyapunov function in order to apply the control theoretic principles. Consider a continuous function $E(\bar{e})$ to be a candidate Lyapunov function as follows:

$$E = \frac{|\bar{e}|^{(\alpha+1)}}{(\alpha+1)} \tag{1}$$

where $\alpha \in (0,1)$ is a user-defined parameter. The second objective is to define the temporal rate of weight as the control input to enforce the stability of the origin $\bar{e} = 0$ as $t \to \infty$. The Lyapunov function in (1) is used to show that it is indeed plausible to achieve this asymptotic stability goal. Taking the temporal derivative of the candidate Lyapunov function (1) produces (by chain rule of differentiation)

$$\frac{\mathrm{d}E}{\mathrm{d}t} = \frac{\mathrm{d}E}{\mathrm{d}\bar{e}} \frac{\mathrm{d}\bar{e}}{\mathrm{d}y} \frac{\mathrm{d}y}{\mathrm{d}z} \frac{\mathrm{d}z}{\mathrm{d}w} \frac{\mathrm{d}w}{\mathrm{d}t}$$

where $\frac{\mathrm{d}E}{\mathrm{d}\bar{e}} \frac{\mathrm{d}\bar{e}}{\mathrm{d}y} \frac{\mathrm{d}y}{\mathrm{d}z} \frac{\mathrm{d}z}{\mathrm{d}w}$ is the weight update for standard gradient descent algorithm and $\frac{\mathrm{d}w}{\mathrm{d}t}$ is the additional term we introduce to the weight update equation. Using (1), we get:

$$\frac{dE}{dt} = |\bar{e}|^{\alpha} \operatorname{sign}(\bar{e}) \left( \frac{e^{-z}}{(1+e^{-z})^2} \right) (x_1 \dot{w}_1 + x_2 \dot{w}_2 + \cdots + x_n \dot{w}_n) \tag{2}$$

Define, $u_1 \triangleq \dot{w}_1, u_2 \triangleq \dot{w}_2, \cdots, u_n \triangleq \dot{w}_n$, and

$$u_i = -k_i \operatorname{sign}(x_i) \operatorname{sign}(\bar{e}) e^z (1 + e^{-z})^2, \quad i = 1, 2, \cdots, n, \tag{3}$$

where $k_i > 0$, for all $i$, are tuning parameters to be chosen by the user. It can be noted that all control inputs $u_1, u_2, \cdots, u_n$ remain bounded due to boundedness assumption of all the inputs $x_i$ and that of $e^z$. Substituting (3) into (2) produces

$$\frac{dE}{dt} = -|\bar{e}|^{\alpha} \left( \sum_{i=1}^{n} k_i |x_i| \right) \tag{4}$$

**Assumption 1.** *At least one input of all $x_i, i = 1, 2, \cdots, n$ is non-zero such that $|x_j| > \gamma > 0$ where $\gamma$ is a priori known scalar for some integers $j \in [1, n]$.*

It can be noted that Assumption 1 is reasonable for many practical applications in that some inputs will always be nonzero with a known lower bound on its magnitude. First main result of the paper is in order.

**Theorem 1.** *Assuming Assumption 1 holds true, let the output of the neural network be given by $y = \sigma(z)$. Let all the inputs $x_i, i = 1, 2, \cdots, n$ be bounded by some a priori known scalar $a \in (0, \infty)$ such that $|x_i| < a$ holds true for all $i$. Then, weight update (3) causes the error $\bar{e} = y - y^*$ to converge to zero in finite time.*

*Proof.* The proof of the theorem is furnished using standard Lyapunov analysis arguments. Consider $E$ defined by (1) as a candidate Lyapunov function. Observing (4), it can be concluded that the right hand side of the temporal derivative of remains negative definite since it involves power terms and norm of inputs. Furthermore, (4) can be rewritten under the assumption 1 as follows:

$$\frac{dE}{dt} \leq -k_{\min} \gamma E^{\beta} \tag{5}$$

where $k_{\min} = \min(k_i), i = 1, 2, \cdots, n$, $\beta = \frac{\alpha}{\alpha+1}$ and $|\bar{e}|^{\alpha} = \left( |\bar{e}|^{\alpha+1} \right)^{\frac{\alpha}{\alpha+1}} = E^{\beta}$ has been utilized. Noting that $E$ is a positive definite function and scalars $k_{\min}$ and $\gamma$ are always positive, the proof is complete by applying (Bhat & Bernstein, 2000, Theorem 4.2). $\square$

## 2.2 Multi Neuron Case

Consider a multi-layer perceptron with $N$ layers where the layers are connected in a feed-forward manner (Bishop, 1995, Chapter 4). Let $x \in \mathbb{R}^n$ be the input to the network where $x = \begin{bmatrix} x_1 & x_2 & \cdots & x_n \end{bmatrix}^\top$ and $|x_i| < c, i = 1, 2, \cdots n$ holds true for some a priori but arbitrary scalar $c \in (0, \infty)$. Let $y \in \mathbb{R}^m$ define the multi-neuron output to the network where $y = \begin{bmatrix} y_1 & y_2 & \cdots & y_m \end{bmatrix}$. Let $y^* \in \mathbb{R}^m$ define the target output values of the network where $y^* = \begin{bmatrix} y_1^* & y_2^* & \cdots & y_m^* \end{bmatrix}$. The

error in the output layer can be expressed as $\bar{e} = [|y_1 - y_1^\star| \quad |y_2 - y_2^\star| \quad \cdots \quad |y_m - y_m^\star|]$. Hence, the scalar candidate Lyapunov function can be written as follows:

$$E = E_1 + \cdots + E_m = \frac{|\bar{e}_1|^{(\alpha+1)}}{(\alpha+1)} + \cdots + \frac{|\bar{e}_m|^{(\alpha+1)}}{(\alpha+1)} \tag{6}$$

As is usually done in the case of feed-forward networks, consider unit $j$ of layer $l$ that computes its output

$$a_j^l = \sum_i w_{ji}^l z_i^l \tag{7}$$

using its inputs $z_i^l$ from layer $l$ where bias parameter has been embedded inside the linear combination and $z_j^l = \sigma(a_j^{l-1})$, where $\sigma$ is non-linear activation function. The aim of this section is to extend the single neuron case to a multi-neuron one. The simplest way do achieve this is to find sensitivity of $E$ to the weight $w_{ji}^l$ of a given layer $l$, which is given by

$$\frac{\partial E_m}{\partial w_{ji}^l} = \frac{\partial E_m}{\partial a_j^l} \frac{\partial a_j^l}{\partial w_{ji}^l} \tag{8}$$

Using standard notation $\delta_j^l \triangleq \frac{\partial E_m}{\partial a_j^l}$ with (7) results in:

$$\frac{\partial E_m}{\partial w_{ji}^l} = \delta_j^l z_i^l \tag{9}$$

It is straightforward to compute $\delta_m^L$ that belongs to the output layer $L$ as shown below:

$$\delta_m^L = \frac{\partial E_m}{\partial a_m^L} = \sigma'(a_m^L) \frac{\partial E_m}{\partial y_m} \tag{10}$$

where $z_m^L$ is replaced by $y_m$ as it is the output layer. Finally, computation of $\delta_j^l$ for all hidden units is given by

$$\delta_j^l = \frac{\partial E_m}{\partial a_j^l} = \sum_k \frac{\partial E_m}{\partial a_k^{l+1}} \frac{\partial a_k^{l+1}}{\partial a_j^l} \tag{11}$$

where units with a label $k$ includes either hidden layer units or an output unit in layer $l + 1$. Combining (7), $z_j^l = \sigma(a_j^l)$ and $\delta_j^l \triangleq \frac{\partial E_m}{\partial a_j^l}$ produces

$$\delta_j^l = \sigma'(a_j^l) \sum_k w_{kj}^{l+1} \delta_k^{l+1} \tag{12}$$

A slightly modified version of assumption 1 is required before the next result of the paper is presented.

**Assumption 2.** *At least one input of all $z_i, i = 1, 2, \cdots, L$ is non-zero such that $|z_n| > \gamma > 0$ where $\gamma$ is a priori known scalar for some integers $n \in [1, L]$ where $L$ is the number of units in layer $l$.*

**Theorem 2.** *Let the weight update for connecting unit $i$ of layer $l$ to unit $j$ of layer $l + 1$ of a multi-layer neural network be given by*

$$\dot{w}_{ji}^l = -k_{ji}^l \operatorname{sign}(\delta_j^l z_i^l) |\delta_j^l z_i^l|^\alpha E^\beta \tag{13}$$

*with some scalar $\beta \in (0, 1)$ such that $\alpha + \beta < 1$ and $k_{ji} > 0$ is a tuning parameter. Then the output vector $y$ converges to $y^*$ in finite time.*

*Proof.* Consider the candidate Lyapunov function $E$ given by (6). The temporal derivative of the Lyapunov function is given by

$$\dot{E} = \sum_m \dot{E}_m = \sum_m \frac{\partial E_m}{\partial w_{jil}} \dot{w}_{ji}^l \tag{14}$$

which can be simplified using (13) and (9) as

$$\dot{E} = -E^\beta \sum_m k_{ji}^l |\delta_j^l z_i^l|^{\alpha+1} \tag{15}$$

Using Assumption 2 it is easy to conclude that for some $k_{\min} = \min\limits_{i,j,l} k_{ji}^l > 0$, the following inequality holds true:

$$\dot{E} \le -k_{\min} \gamma^{\alpha+1} E^\beta \tag{16}$$

Noting that $E$ is a positive definite function and scalars $k_{\min}$, $\gamma$ are always positive, the proof is complete by applying (Bhat & Bernstein, 2000, Theorem 4.2). $\qquad\square$

**Remark 1.** *It can be seen that weight update (13) (or respectively (3)) is in a feedback control form where state $z_i$ and $\delta_j$ (respectively $x_i$ and $\bar{e}$) are being used for influencing the learning process.*

### 2.3 SETTLING TIME FOR NEURAL NETWORK TRAINING

Theorem 1 and 2 prove that using candidate Lyapunov function as the loss function and the temporal rate of change of the loss function as the weight update equation, we can convert supervised learning framework into a control problem. (Bhat & Bernstein, 2000, Theorem 4.2) states that if there is continuous function that is positive definite and it's rate of change with respect to time is negative definite, then it's finite time stable equilibrium is at origin. The theorem also gives the settling time when the above conditions are satisfied by the control system. For our case, this means that the Lyapunov based loss function will converge in finite time and time taken to converge is as given below:

**The settling time for Single Neuron case:**

$$T \le \frac{1}{k_{min}\gamma(1-\beta)} E_{ini}^{(1-\beta)} \tag{17}$$

**The settling time for Multi Neuron case:**

$$T \le \frac{1}{k_{min}\gamma^{\alpha+1}(1-\beta)} E_{ini}^{(1-\beta)} \tag{18}$$

where $\gamma$ is the lower bound on the inputs, $k_{min}$ is the minimum value of tuning parameter $k$, $\alpha$ is the scalar used in the Lyapunov loss function and $E_{ini}$ is the initial value of loss, i.e. loss at t = 0.

### 2.4 SENSITIVITY TO PERTURBATIONS

This section considers the robustness of training a neural network based on our proposed algorithm. In control theoretic terms, perturbation can be understood either as a disturbance to the process or as modelling uncertainty. When supervised learning is viewed as a control process, perturbation in each neuron of the input layer can be seen as external noise. This perturbation could cause the control system to diverge. In this section, we develop theoretical claims that even with perturbed inputs, training a neural network with the proposed algorithm will converge. The following assumption of the upper bound on the perturbation of inputs is invoked.

**Assumption 3.** *There exists an a priori known constant $M > 0$ such that all inputs $x_i, i = 1, 2, \cdots, N$ admit additive perturbations $\Delta x_i$ such that*

$$|\Delta x_i| \le M|x_i|^\alpha \tag{19}$$

*for all $i$ where $N$ is the number of inputs.*

The following result is in order.

**Theorem 3.** *Let assumptions 2 and 3 hold true. Let the weight update for connecting unit $i$ of layer $l$ to unit $j$ of layer $l + 1$ of a multi-layer neural network be given by (13). Then the output vector $y$ converges to $y^*$ in finite time in the presence of additive perturbations $\Delta x_n, n \in [1, L]$ if $k_{\min} > M$.*

*Proof.* It can be seen that the perturbation is considered only in inputs. Hence all hidden layer weights are updated as done in the proof of Theorem 2. Hence, the dynamics of learning results in the following revised temporal derivative of Lyapunov function:

$$\dot{E} = -E^\beta \sum_m k_{ji}|\delta_j z_i|^{\alpha+1} + \sum_{p=0}^n k_{1p}|\delta_p x_p|^\alpha \operatorname{sign}(\delta_p x_p)\delta_p \Delta x_p \tag{20}$$

where $k_{1p}$ is the gain parameter for training of all the input layer weights. The expression in (20) can be simplified using Assumption 3 as follows:

$$
\begin{aligned}
\dot{E} &\leq -E^\beta \sum_m k_{ji}|\delta_j z_i|^{\alpha+1} + E^\beta \sum_{p=0}^n k_{1p}|\delta_p x_p|^{\alpha+1} M, \\
&\leq -E^\beta \sum_m (k_{ji} - M)|\delta_j z_i|^{\alpha+1}
\end{aligned}
\tag{21}
$$

where inputs $z_i$ now collect inputs $x_i$ as well. Similar to the proof of Theorem 2, (21) can be re-written by applying Assumption 2 as follows:

$$\dot{E} \leq -(k_{\min} - M)\gamma E^\beta \tag{22}$$

Since $k_{\min} > M$, the proof is complete by applying (Bhat & Bernstein, 2000, Theorem 4.2). $\qquad\square$

## 3 EXPERIMENTS

This section presents empirical evidence that the theoretical results derived in the above sections hold for real-life applications.

**Convergence Rate for training.** To analyze the training convergence rate of the proposed method, we train a multi-layer perceptron (covered in Section 2.2) to perform regression on the Boston Housing dataset (Harrison Jr & Rubinfeld (1978)). Experiment on single neuron case is covered in Appendix A.2.2

**Training details.** We compare our proposed method (Lyapunov based loss and modified weight update equation) with traditional $L_1$, $L_2$ loss functions and standard SGD weight update equation. The modifications to the weight update equation for Lyapunov loss are as stated in (15) for multi-neuron case. We plot the training and test loss with respect to time in Figure 1.

The details about the training examples, epochs, learning rate etc. are specified under the description of the figures. We observe that the Lyapunov Loss function converges faster than $L_1$ and $L_2$ loss functions, demonstrating the results proven extensively in Section 2. This is attributed mainly to the non-Lipschitz weight updates given by (3) and (13).

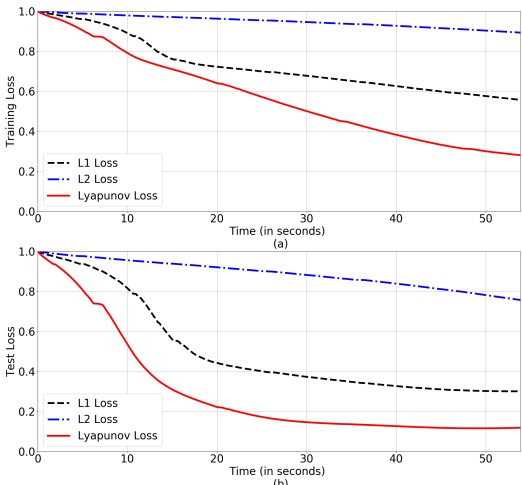

The settling time as well as metrics of the trained networks are reported in Table 1. The time taken by the proposed algorithm to converge falls within the a priori upper bound. We also observe that the metrics (Accuracy for single neuron case and rmse (Root mean square error) for multi-neuron case) achieved by the proposed method is better than the baselines. Admittedly, the theoretical upper bound on settling time for single neuron and multi-layer perceptron produced by (5) and (16) are very conservative, yet it is an a priori deterministic upper bound and the aggressive learning proves to be faster than traditional loss functions in the scenarios considered here.

Figure 1: Comparison of convergence with respect to time for the multi-layer perceptron trained on the Boston Housing dataset. All networks are trained for 4000 epochs with learning rate = 0.02 and $\alpha = 0.68$.

| Experiment | Theoretical Upper Bound (in seconds) | Exp. Convergence Time (in seconds) | | | Metric | | |
|---|---|---|---|---|---|---|---|
| | | $L_1$ | $L_2$ | $Lyap.$ | $L_1$ | $L_2$ | $Lyap.$ |
| SingleNeuron (acc) | $\sim 3.4e3$ | 0.065 | 0.064 | 0.025 | 0.95 | 0.95 | 1.0 |
| MLP (rmse) | $\sim 1.4e10$ | $4.3e3$ | $3.4e3$ | $3.1e3$ | 0.091 | 0.179 | 0.085 |

Table 1: Settling time in seconds for each experiment conducted. We compare the time taken for convergence by three different loss functions, $L_1$, $L_2$ and Lyapunov Loss function. The training conditions were similar for individual cases in the experiment. The single neuron case is trained on Iris dataset and the metric used is Accuracy (higher is better). For Multi-layer perceptron, we use Boston Housing dataset and the metric used for this is rmse (lower is better).

This can be attributed to the fact that we operate in the realm of non-smooth functions which tend to be more aggressive than the smooth $L_2$ function we usually encounter in most learning problems. We can clearly observe that the loss value does not converge to zero. According to (Bhat & Bernstein, 2000, Theorem 5.2), a control system will converge to a non-zero value at steady state, if there is constant perturbation in the input. Usually in real life datasets, there is an inherent bias or constant perturbation, which prevents the loss to converge to zero. Of course, by setting $\alpha = 0$, we can reject all persisting disturbances (Orlov (2005)) and force the loss to converge to zero, but this may result in discontinuity in back propagation (discussed in Appendix A.1) and result into large numerical errors.

**Hyperparameter analysis and its effect on training.** In this section, we attempt to give an empirical analysis as to how changes in the hyperparameters like $k$ and $\alpha$ affect the training of the neural network. In order to study the effect of tuning $k$, we work with the single neuron case for simplicity and ease of understanding. We use Iris dataset for this experiment Dua & Graff (2017). Existing literature in control theory suggests that a monotonic increase in the tuning parameter, $k$ should result in a monotonic decrease in the corresponding loss. In neural network terminology, $k$ can be considered as the 'learning rate'. We can clearly see in Figure 2 that $k$ behaves like learning rate parameter where increasing $k$ makes the learning more aggressive.

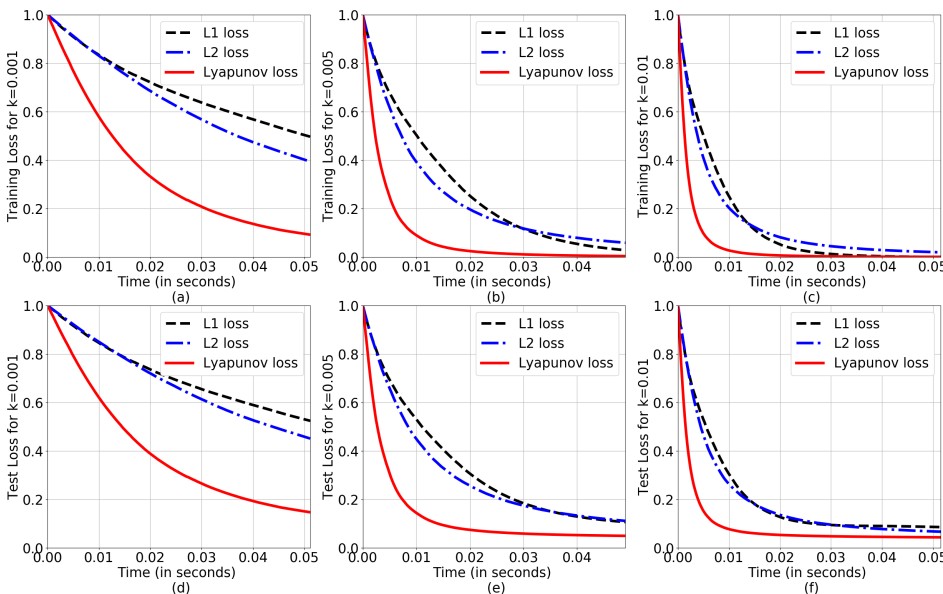

Figure 2: Comparison of training and test convergence with respect to time for different values of tuning parameter, $k$ in a single neuron trained on the Iris dataset. We convert the problem into a binary classification problem by considering only two output classes. All networks are trained for 600 epochs with 80 training examples and 20 test examples. The learning rate for all $L_1$, $L_2$ and Lyapunov is set to the tuning parameter value $k$ and $\alpha = 0.8$

From Table 2, we observe that the settling time for the training loss of our Lyapunov loss function is always less than $L_1$ and $L_2$. We also observe that the Accuracy of our method is either better

or similar to the baselines, which shows that the optimized solution achieved by our method is either better or on par with the baselines. For the multi-neuron case, we have defined $k_{ij}^l$ for every weight. Similar to adaptive learning rate, we can either tune $k_{ij}^l$ for every weight (which can be quite cumbersome for larger neural networks) or we can set it arbitrarily to the same value for all weights. It must be noted that $k_{ij}^l$ can be set to any value greater than an a priori known positive constant $M$.

| Experiment for single neuron case on Iris dataset | Theoretical Upper Bound (in seconds) | Exp. Convergence Time (in seconds) | | | Accuracy on Test Set | | |
|---|---|---|---|---|---|---|---|
| | | $L_1$ | $L_2$ | $Lyap.$ | $L_1$ | $L_2$ | $Lyap.$ |
| $k = 0.001$ | $\sim 37213.848$ | 0.312 | 0.310 | 0.0918 | 0.6 | 0.95 | 0.95 |
| $k = 0.005$ | $\sim 7442.769$ | 0.0943 | 0.0942 | 0.0344 | 0.95 | 1.0 | 1.0 |
| $k = 0.01$ | $\sim 3721.385$ | 0.0566 | 0.0573 | 0.0221 | 0.95 | 1.0 | 1.0 |

Table 2: Settling time in seconds for different values of tuning parameter, $k$ for the single neuron case on Iris dataset. We compare the time taken for convergence by three different loss functions, $L_1$, $L_2$ and Lyapunov Loss function. The training conditions were similar for individual cases in the experiment.

We will discuss the impact of different values of $\alpha$ on the training process of our neural network. For the given experiments, we tested the network for various values of $\alpha$ ranging between $(0, 1)$. We observed that the training loss follows equations proved in above sections as long as $\alpha \in (0.5, 0.9)$. If $\alpha$ happens to be too close to zero, the resulting control law becomes discontinuous thereby resulting in numerical instabilities owing to the fact that the current ODE solvers are unable to handle functions that are discontinuous. More details about the case $\alpha = 0$ is given in Appendix A.1 (Bhat & Bernstein, 2000, Theorem 5.2).

**Stability of learning in the presence of amplitude-bounded perturbations.**

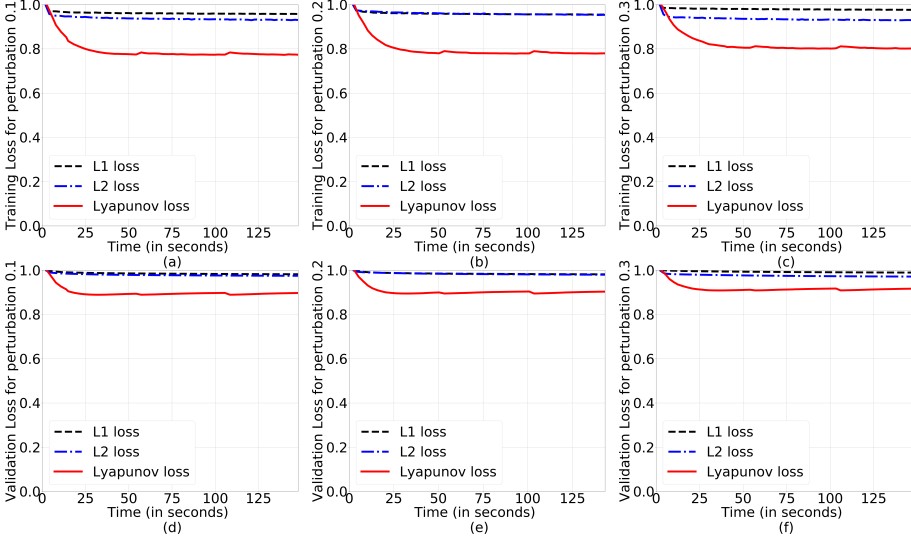

Figure 3: Comparison of training loss convergence with respect to time for different values of input perturbations, $\Delta x$ for a multi-layer perceptron trained on the IMDB Wiki Faces dataset.The a priori upper bound on input perturbations are as follows: (a) $\Delta x = 0.1$, (b) $\Delta x = 0.2$, (c) $\Delta x = 0.3$ Learning rate/$k = 0.0009$, $\alpha = 0.8$, epochs=100

In this section, we present results that demonstrate the robustness of proposed algorithm to bounded perturbations in the input while maintaining convergence of training dynamics. For this case, we train a multi-layer perceptron on a regression task of predicting the age given the image of a face. We use IMDB Wiki Faces Dataset Rothe et al. (2015) which has over 0.5 million face images of celebrities with their age and gender labels. Only part of the dataset is used for this experiment, i.e. 20,000 images for training and 4,000 images each for validation and test.

For adding input perturbations, we asssume that an a priori upper bound, $M$ is known on the amplitude of possible input perturbations following Assumption 3. We add input perturbations to each pixel in the image using a randomized uniform distribution ranging from $(-\Delta x, \Delta x)$. Hence, our additive noise remains in the abovementioned a priori bounds. We vary the values of $M$ from $(0.1, 0.3)$ with a $0.1$ increment, giving us three training cases. Figure 3 shows that the proposed training loss still manages to achieve steady state error and does not diverge due to perturbations introduced in the input dataset. Since we are dealing with noisy data, the loss values converge to a non-zero value in the steady-state

**Experiments on larger datasets.** In this section, we present results to demonstrate the performance of our proposed algorithm on a larger dataset. We use the entire IMDB Wiki Faces Dataset Rothe et al. (2015) with 0.5 million images for this experiment. Training dataset consists of $\sim$0.2 million images and the test and validation set consists of $\sim$0.1 million images each. As described in the previous section, a multi-layer perceptron is trained to predict the age from the image of a face. The MLP is trained for 100 epochs with learning rate 0.0005 for all three loss functions and with $\alpha = 0.7$ for Lyapunov loss function. From Figure 4 and Table 3, we can see that our proposed algorithm achieves similar generalization / testing rmse (Root mean square error) as $L_1$ and $L_2$ baselines while providing finite time convergence guarantees even on large datasets. The convergence time is also within the theoretically derived upper bound.

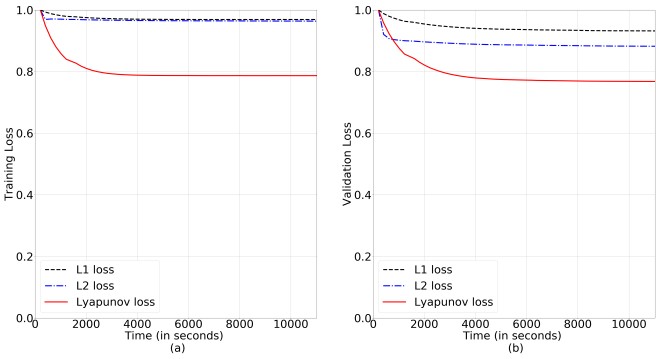

Figure 4: Comparison of training convergence with respect to time for 0.5 million IMDB Wiki dataset

| Experiment | Theoretical Upper Bound | Exp. Convergence Time (in seconds) | | | Metric rmse | | |
|---|---|---|---|---|---|---|---|
| | (in seconds) | $L_1$ | $L_2$ | $Lyap.$ | $L_1$ | $L_2$ | $Lyap.$ |
| IMDB Wiki | $8.3e6$ | 980.40 | 712.99 | 467.96 | 0.414 | 0.415 | 0.416 |

Table 3: rmse on IMDB Wiki test dataset for $L_1$, $L_2$ and Lyapunov Loss function.

## 4 CONCLUSION AND FUTURE WORK

This paper studies the training of a deep neural network from control theory perspective. We pose the supervised learning problem as a control problem by jointly designing loss function as Lyapunov function and weight update as temporal derivative of the Lyapunov function. Control theory principles are then applied to provide guarantees on finite time convergence and settling time of the neural network. Through experiments on benchmark datasets, our proposed method converges within the a priori bounds derived from theory. It is also observed that in some cases our method enforces faster convergence as compared to standard $L_1$ and $L_2$ loss functions. We also prove that our method is robust to any perturbations in the input and convergence guarantees still hold true. The given a priori guarantees for the convergence time is a desirable result for training networks that are extremely difficult to converge, specifically in Reinforcement Learning. A future scope of this work may be to convert the continuous time analysis framework to discrete time. This study introduces a novel perspective of viewing neural networks as control systems and opens up the field of machine learning research to a plethora of new results that can be derived from control theory.

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

## A  APPENDIX

### A.1  IMPACT OF $\alpha$ ON TRAINING

The case when $\alpha = 0$, as depicted in Figure 5, raises continuity issues.

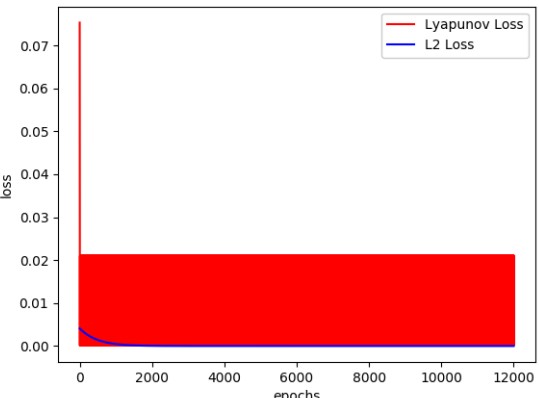

Figure 5: Depiction of the numerical instability observed when we take $\alpha = 0$. Effectively, at this point, the loss function contains a discontinuous signum function.

Consider the left and right limits of the function $f(\varrho) = |\varrho|^\alpha \text{sign}(\varrho)$, where $\varrho = \delta_j z_i$ as appearing in (15). It can be seen that $\lim_{\varrho \to 0^-} f(\varrho) = \lim_{\varrho \to 0^+} f(\varrho) = 0$ for $\alpha \in (0, 1)$. However, for $\alpha = 0$, $\lim_{\varrho \to 0^-} f(\varrho) = -1$ and $\lim_{\varrho \to 0^+} f(\varrho) = 1$. It should be noted that function $f(\varrho)$ is non-Lipschitz since $\partial f / \partial \varrho$ tends to infinity in the limit $\varrho \to 0$. We agree that $\alpha = 0$ reduces the loss function to L1 loss, but the control update becomes purely discontinuous due to the presence of $f(\varrho)$ in (15). We do not deal with this case for the reasons of continuity as mentioned in the main paper.

### A.2  EXPERIMENTS

### A.2.1  EXPERIMENTAL SETUP

For our experiments, we use the NVIDIA GEFORCE GTX1080Ti GPU card with 12GB RAM, 4-core CPU with 32GB of RAM for training. The training code is in Python. All networks are trained on data with an $80\% - 20\%$ train data-test data split.

### A.2.2 CONVERGENCE EXPERIMENTS ON SINGLE NEURON

In this experiment, we perform binary classification on the Iris dataset (Dua & Graff (2017)) as a representative example of single neuron case presented in Section 2.1.

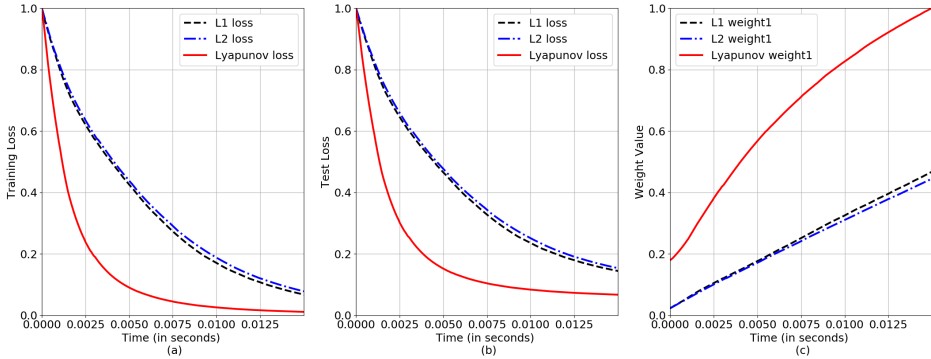

Figure 6: Comparison of convergence with respect to time for a single neuron trained on the Iris dataset. We convert the problem into a binary classification problem by considering only two of the three output classes. All networks are trained for 2100 epochs with 80 training examples and 20 test examples, $\alpha = 0.8$, learning rate / k = 0.01

From the Figure 6, we can clearly see that Lyapunov loss function with control weight update converges much faster towards zero as compared to $L_1$ and $L_2$ loss with standard gradient descent weight update. This shows that explicitly adding a control weight update (15) that drives the loss towards zero is helpful in reaching convergence faster.

### A.2.3 BOSTON HOUSING DATASET EXPERIMENT WITH PERTURBATIONS

The Boston Housing Dataset predicts the price of houses in various areas of Boston Mass, given 14 different relevant attributes, like per capita crime rate and pupil-teacher ratio by town. The 506 examples in the data are divided into a 80-20 training-test split to give us 505 training examples and 101 test examples respectively. We consider three cases here, $M = 0.1$, $M = 0.2$, and $M = 0.3$. Table 4 presents the theoretical upper bounds for the proposed Lyapunov function's settling time and experimental results for the training conducted for different upper bounds assumed for the input perturbation. We observe that the settling time for the proposed Lyapunov function is similar to the one observed for $L_2$ loss function whereas it performs way better than the $L_1$ loss function.

| Experiment for MLP case on Boston dataset | Theoretical Upper Bound (in seconds) | Exp. Convergence Time to within 10e-9 (in seconds) | | | Metric rmse | | |
|---|---|---|---|---|---|---|---|
| | | $L_1$ | $L_2$ | $Lyap.$ | $L_1$ | $L_2$ | $Lyap.$ |
| $M = 0.1$ | $\sim 1.97e^{11}$ | 1277.66 | 906.61 | 755.59 | 0.095 | 0.145 | 0.092 |
| $M = 0.2$ | $\sim 5.67e^{10}$ | 1297.11 | 1097.62 | 914.78 | 0.096 | 0.146 | 0.093 |
| $M = 0.3$ | $\sim 2.73e^{10}$ | 1290.48 | 1197.53 | 998.05 | 0.093 | 0.147 | 0.101 |

Table 4: Settling time in seconds for different values of the upper bound on additive input perturbations, $M$ for the multi layer perceptron case on Boston Housing dataset. We compare the time taken for convergence by three different loss functions, $L_1$, $L_2$ and Lyapunov Loss function. The training conditions were similar for individual cases in the experiment.

### A.2.4   IMDB WIKI FACES EXPERIMENT ON PERTURBATIONS

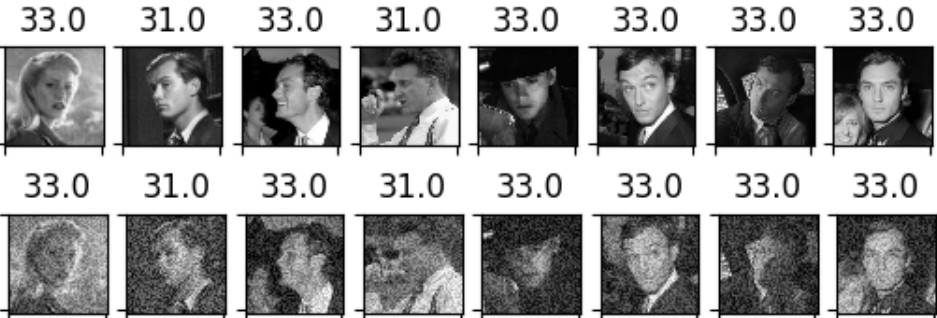

Figure 7: Pictorial representation of the effect of random additive noise with an upper bound of $0.2$ on IMDB Wiki Faces Dataset. In this experiment, we work with monochrome images for the sake of simplicity.

Figure 7 shows us how the noise affects the training images. We take the specific case where $M = 0.2$. We can clearly observe that the images obtained after adding input perturbations happens to be quite noisy. Figure 8 and Figure 9 show that even when inputs are perturbed, our proposed method converges in finite time.

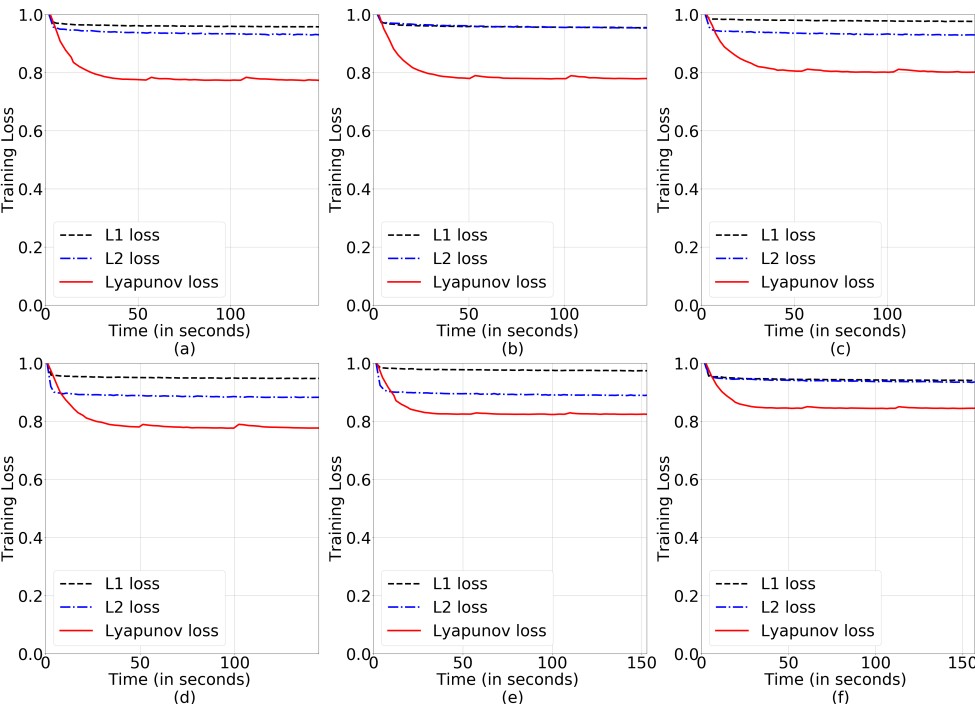

Figure 8: Comparison of training loss convergence with respect to time for different values of input perturbations, $\Delta x$ for a multi-layer perceptron trained on the IMDB Wiki Faces dataset. The a priori upper bound on input perturbations are as follows: (a) $\Delta x = 0.1$, (b) $\Delta x = 0.2$, (c)$\Delta x = 0.3$, (d) $\Delta x = 0.4$, (e) $\Delta x = 0.5$, and (f) $\Delta x = 1.2$.

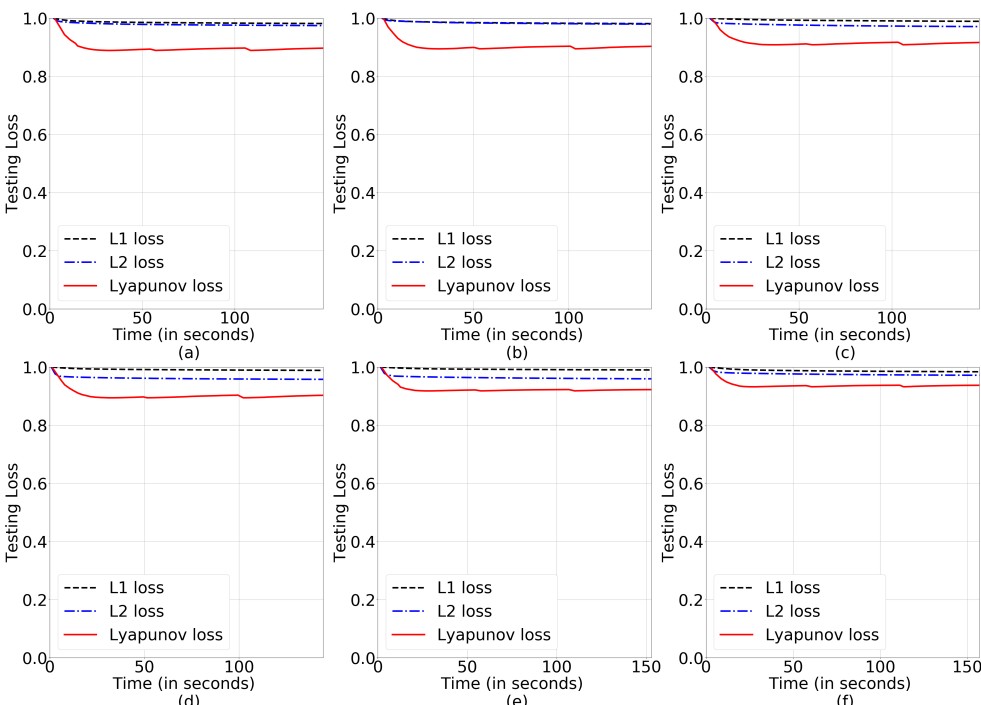

Figure 9: Comparison of test loss convergence with respect to time for different values of input perturbations, $\Delta x$ for a multi-layer perceptron trained on the IMDB Wiki Faces dataset.The a priori upper bound on input perturbations are as follows: (a) $\Delta x = 0.1$, (b) $\Delta x = 0.2$, (c)$\Delta x = 0.3$, (d) $\Delta x = 0.4$, (e) $\Delta x = 0.5$, and (f) $\Delta x = 1.2$.

