# OpenReview forum: "A priori guarantees of finite-time convergence for Deep Neural Networks"
_ICLR.cc/2021/Conference — Reject_

### Official Review · AnonReviewer3 · 2020-10-26
**Using Lyapunov function to model training is interesting, but scalability of the results is an issue.**

**Rating:** 4
**Confidence:** 5

**Review:**

The authors in this paper make an attempt in providing finite time convergence guarantees of the training process of neural networks, using ideas from control theory.  The loss function in the training process is framed as a Lyapunov function.  The training process at each time step is seen as  assigning dynamics to the Lyapunov function over time. The convergence of which can then be analyzed using standard control theoretic techniques.

For some fixed input - output target, the idea is to come up with a weight update rule which guarantees the convergence rate with some assumptions on the inputs. This  is the novelty in the paper. The extension to multi-layer case is  an extension of the back propagation algorithm.

Though the above is an interesting contribution in itself,  I am not convinced that the results for a fixed input case would generalize well to the batched input case. Which in my opinion is more general,  and has enabled the training of large scale neural networks. The authors have analyzed the robustness to perturbations  in Section 2.4. Specifically Eqn 22. Where the authors have bounds on the perturbation limits, under which it can still guarantee convergence rates. For the batched case it might need some restrictions on the choice of samples in a batch.

The next concern I have is the experiments definitely looks very insufficient. The current experiments include much smaller datasets. I would be interested to see how this technique performs on some of the larger  neural network architectures. Since convergence guarantees become more important, only when the time it takes to train a network is much longer.

---

> ### Author Response · Authors · 2020-11-19
> **Response: Using Lyapunov function to model training is interesting, but scalability of the results is an issue.**
>
> Thank you for your time to read and review our manuscript.
> The assumption that we have used to derive our theorems is that atleast one of the input dimensions should be greater than zero and the all input dimension values should be finite i.e. less than some scalar value ‘a’. While training any machine learning task, the inputs are usually normalized (ranges from -1 to 1 or 0 to 1), hence the assumption does not seem unreasonable for any machine learning application. This assumption holds true even in batched case, as the inputs will be normalized to some finite range. Thus, our algorithm does not depend or have any restrictions on choice of samples in a batch.
> We include experimental results (test rmse, error plots) for larger dataset (0.5 million images) in the modified submission on Page 9. The convergence bounds and experimental convergence time will be updated in the table in a couple of days.

---

> > ### Author Response · Authors · 2020-11-23
> > **Response: Using Lyapunov function to model training is interesting, but scalability of the results is an issue.**
> >
> > We have updated Table 3 with the convergence bounds and experimental convergence time on larger dataset (0.5 million images).

---

### Official Review · AnonReviewer4 · 2020-10-27

**Rating:** 4
**Confidence:** 3

**Review:**

This work studies the finite-time convergence for neural networks. In particular, it tries to recast the problem of training neural networks as a control problem. Supervised learning is then reformulated as a non-linear control problem with a Lyapunov based loss. The weight update is then transformed to be the control inputs. Finally, convergence results are obtained with standard theory from non-linear systems.

Overall, connecting neural networks with classical control theory is an interesting direction. However, results presented in this paper seems limited, and it is not clear what contributions the current work really bring to the community.
(1) there does not seem to be enough innovation in this paper. To me, the result simply follows from the classical control theory. The authors simply try to mimic the theory by having a candidate Lyapunov loss and continuous weight update equations. It is not clear why these are used for neural networks at the first place; rather, it seems that these are only applied for the sake of proving some technical results. For example, does the candidate Lyapunov loss actually generalize better (theoretically or empirically)? what's the property of it? Howe does it compare to traditional loss function? It is not convincing for me why someone should use it for training neural networks. It seems to be an artifact used solely for the theorem.
(2) relatedly, the experiments focus on plotting the training loss of Lyapunov loss and l1/l2 loss. From my perspective, this is not informative. The loss function is likely not on the same scale; it is probably better to plot a "normalized" version so that the comparison indeed makes sense. Again, such a comparison does not reveal any interesting property about the new loss, e.g.., generalization/testing error?
(3) Please clarity to what extent, the results hold with respect to different activation function. In Section 2.1, it is explicitly mentioned that sigmoid activation is used. In Section 2.2, the authors use the same notation \sigma. Does the result hold for other common activation functions? If not, any comments on the difficulties?

---

> ### Author Response · Authors · 2020-11-19
> **Response: Review**
>
> Thank you for your time to read and review our manuscript.
> The main innovation in this paper is to model neural networks as a dynamical control system. Recasting the problem of supervised learning as a dynamical control problem has several benefits. For example, it becomes possible to compute a priori convergence bounds, simpler hyper-parameter optimization and a finite-time convergent weight update. The main argument of the paper is not to use a particular loss function, but to demonstrate that by treating the loss function as a Lyapunov function and modifying the weight update accordingly, the supervised learning framework can benefit from the well-known methods of dynamical control systems. As for empirical generalization of the Lyapunov loss, please see the comparison of test error plots of the proposed Lyapunov loss function and that of traditional loss functions in the revised manuscript on page 6, 7, 8 and 9.
> Thank you for your comments on the loss plots. We have modified the plots so that they are normalized by the respective maximum of the loss function (which happens to be at the initial condition). We have also included these normalized plots of test error in the revised manuscript. We believe the information conveyed by such a comparison is the rate at which the proposed weight update converges. This has direct implication to the learning problem.
> The results hold for all activation functions which are once differentiable. We mentioned a particular activation function so as to show detailed derivation with backpropagation for the weight update equation. Looking at equation (8) of the manuscript, it is a requirement to have the activation function σ such that the partial derivative with respect to weight w exists. All such activation functions are admitted by the proposed theory.

---

### Official Review · AnonReviewer1 · 2020-10-27
**Interesting connections between neural network training and control, as well as control-theoretic analysis of the neural network loss function**

**Rating:** 7
**Confidence:** 3

**Review:**

This paper presents a Lyapunov based analysis of the loss function in neural network training and derives a priori upper bounds on the settling time of the training, which somewhat complements existing studies. The supervised neural network learning problem is formulated as a control problem with the weight parameters being the control input, and the learning problem as a tracking problem. Analytic formula for computing the finite-time upper bound on the settling time is provided under suitable assumptions on the input. Furthermore, the loss function is also shown robust against input perturbations.

This paper contains some interesting ideas in revealing relationships between control and neural network learning, which is a plus. Hopefully, this can further motivate exploration and application of more control-theoretic tools to understanding of neural network training. Although this paper is fairly readable, the presentation and organization can be improved. Several detailed comments are provided below.

i) In control, particularly tracking problem, it is known that there is a given reference signal y(t) one wants the control plant to track. Nonetheless, here in the discussion the y^\ast I guess is determined by the loss function, training method, data, as well as the neural network architecture altogether, right? What exactly is this y^\ast? How is it related to e.g., the equilibrium point of the weight parameters and stationary points in the optimization context?

ii) The current analysis in Section 2 pertains to a single data point? How would having more data points affect the analysis and the results? In that case, what would be the y^\ast? Or will be functions of the input?

iii) In the experiments, since the Lyapunov loss function and other loss functions are plotted, how does the loss function convergence correspond to the learned weights parameters? In the context learning, one is more interested in the neural network parameters that not only capture the training data but also predict well the unseen ones? So it would also be interesting to present the corresponding testing results?

---

> ### Author Response · Authors · 2020-11-19
> **Response:  Interesting connections between neural network training and control, as well as control-theoretic analysis of the neural network loss function**
>
> Thank you for your time to read and review our manuscript.
> i) The target output is denoted by $y^{\ast}$ for a given supervised learning task. The output of the neural network is y. The weight updates are such that y tracks $y^{\ast}$ in finite time. In an analogy with control systems, y stands for the output of the system and $y^{\ast}$ the commanded signal. In optimization, we try to minimize a cost / performance criteria where the weight update is obtained using equation (13). In our control theory based formulation, we have treated the cost / loss as a Lyapunov function. Then, the weight update is treated as a control signal for the plant (neural network) such that the temporal derivative of the Lyapunov function is always negative definite. Hence, the optimization is achieved via proper control synthesis.
> ii) The analysis in Section 2.1 corresponds to the case of a single neuron. The motivation behind providing that analysis is to demonstrate the main concept of finite-time convergent learning. This motivates the development of a more complex multi-neuron case.  Of course, the single neuron case is not practically useful.
>  Both single and multi neuron case analysis hold for multiple data points. Theoretically, both theorems admit multiple data points. Empirically, we demonstrate this for the single neuron case in section 3 Figure 2, where we trained the single neuron case with Iris dataset (80 data points in training). For multi-neuron case, we use larger datasets and show that the analysis works for multiple data points (Page 9, figure 4). In case we have misunderstood what was meant by a single datapoint (we assume it means one input, output pair in the dataset), we welcome the reviewer to clear our misunderstanding.
> iii) We have included the corresponding test results in the modified submission in Figure 1, 2, and 3.

---

### Official Review · AnonReviewer2 · 2020-10-30
**A priori guarantees of finite-time convergence for Deep Neural Networks**

**Rating:** 7
**Confidence:** 3

**Review:**

The paper aims to make strides towards a theoretical understanding of Deep neural networks, which remains elusive to date. This paper uses a control theoretic formulation to analyze the convergence rate of deep neural networks. More specifically, a Lyapunov based analysis of the loss function is used to derive a priori upper bound on the settling time of a restricted set of fully connected neural network architectures with some assumptions on the input space.

I'm interested to know, for what kind of real-world tasks or datasets is their assumption on the boundedness of the input valid?
Although the proposed Lyapunov loss provides the possibility of analyzing convergence guarantees a-priori, how does this affect the performance of the underlying model on test data?

The paper provides experiments supporting their theoretical claims for MLPs on a regression task and a single neuron on a classification task. They show that their proposed Lyapunov loss converges faster than the L1 and L2 losses, and faster than the a-priori upper bound. Can a similar loss function for MLPs on classification tasks be easily derived? In other words, do these results easily extend to classification tasks?

And what effect does the new loss have on overfitting?

I'm a bit confused by the theoretical upper bound. The derived upper bounds in Table 1 are orders of magnitude higher than the actual time taken, even with the L1 and L2 losses. What does this mean? What's the use of the upper bound in this case?

---

> ### Author Response · Authors · 2020-11-19
> **Response: A priori guarantees of finite-time convergence for Deep Neural Networks**
>
> Thank you for your time to read and review our manuscript.
>  Most of the real world datasets or tasks have a defined range of input values. For example, image values range from 0 to 255. Also, when the input is provided to the network, it is usually normalized, hence the boundedness of assumption holds true for a wide range of tasks and datasets.
> We have included graphs on test data and accuracy in the revised manuscript.
> Regarding classification tasks, yes these results can be extended to classification tasks as well. However, it is needed to identify suitable Lyapunov functions that result in a continuous finite-time update. Certainly, more work is needed before the presented results become applicable for classification tasks.
> From a control theoretic perspective, a more aggressive controller will usually produce a large overshoot while tracking a command input. In the present case, the weight update is being treated as the controller. Hence, setting the gains $k_{ij}$ large will result in oversensitive training. Setting ⍺≪1 close to zero will also result in a highly non-Lipschitz training update. These scenarios correspond to overfitting. This can be seen, for example, in the limiting case (not covered by theory) in appendix A where ⍺=0 causes the oscillations in training update due to limitations of explicit Euler discretization. It is therefore recommended to have a judicious balance between the required convergence rates and the ensuing overfitting of the network.
>
> We agree that the bounds are conservative in nature. This is due to the fact that the upper bound on settling time is computed using Lyapunov derivative. In future work, we plan to show how to derive less conservative bounds.

---

### Public Comment · ~Mouhacine_Benosman1 · 2020-11-12
**The results of finite-time optimization using signed first order flows and Lyapunov function costs are not novel, sorry !**

This is a nice attempt to apply results of continuous optimization in finite-time to the case of DNNs’ training.

However, the authors are invited to compare this work to the recent work Romero et al . ICLM 2020 ( https://proceedings.icml.cc/static/paper_files/icml/2020/4879-Paper.pdf) on the subject of continuous finite-time optimization. Indeed, it appears to me that the Lyapunov cost that the authors are arguing to be novel is simply the Lyapunov function used in this ICML paper (see Proof of Theorem 1 sketch ), where the function $f$ is replaced with the output of the DNN in this particular case. The optimization flow its self is very similar to the signed flow introduced in this ICML paper, referred to as q-SGF, leading to similar finite-time convergence results.

Besides, it also appears to me that the theoretical analysis in this submission is incorrect due to the potential discontinuity of the optimization flow. Indeed, the authors are clearly stating that the acceleration (in continuous time) observed numerically is due the ‘aggressive’ discontinuous flow. Well, that might be true, but that discontinuity needs to be carefully studied, since the argument that the authors are using in their Lyapunov analysis is only valid for Lipschitz continuous flows. For discontinuous flows, one must use the notion of differential inclusion for example, and the associated Lyapunov theory, please refer to the supplementary material of the ICML paper cited above (can also be found in the more general version of the work, which includes the case of time varying cost functions at  https://www.merl.com/publications/docs/TR2020-088.pdf ).

Finally, it seems to me that the notion of continuous time optimization is rather superfluous in the context of DNNs due to their large scale. Indeed, one cannot expect to use a stiff ODE solver to be able to solve the discontinuous flows with the very high dimensions associated with DNNs. As such this work is not suitable for such application, and a proper discretization scheme is needed for that. The catch is, it is far from proven that any explicit discretization will lead to the same finite-time convergence result.

---

> ### Author Response · Authors · 2020-11-12
> **Discussion on novelty and discretization**
>
> Thank you for a detailed reading and review of our work. We appreciate your comments.
>
> As for the title of the comment, indeed, the very idea of Lyapunov functions for continuous finite-time optimization is not new. We do not intend to claim it as our contribution. We also agree that we have not identified a new Lyapunov function. We have stated on the second page that our focus is on posing the learning problem as a control theoretic problem where existing Lyapunov theory is utilized. Perhaps the novelty claim made in Introduction is misleading and we agree to change it as follows: “The novelty lies in the fact that the weight update is cast as a finite-time control synthesis such that the loss function is proven to be a valid Lyapunov function”. Thank you for this comment.
>
> However, we do stand by our claim that Deep Neural Networks have not been rigorously studied from a control theory perspective. In fact, the focus of the ICML 2020 paper being cited in your comment is on continuous and discontinuous differential equations and inclusions arising during optimization of cost functions. The main focus also seems to be on discretization. Theoretically, the discontinuous case in our results arises only when alpha=0 when the differential inclusion has to be considered in the sense of Filippov’s definition . We would like to stress that our theorems do not cover this case. The solutions of the differential equations are understood as defined in Bhat and Bernstein (SIAM, 2000). Of course, with very small values of 0<alpha<1, it is well known that the right hand side of the differential equation becomes non-Lipschitz and numerical methods may not give a solution that matches the corresponding analytical one especially in the presence of disturbances. Hence, we do not understand the comment why our results are incorrect for the case when alpha is nonzero.
>
> As for the comment on superfluousness of continuous time optimization, we would like to point out a few references that help bridge the divide between the continuous and discrete-time analysis that the comment alludes to. We would also like to point out the latest advances in discretization algorithms for discontinuous cases (alpha=0). The implicit numerical schemes reported by Vincent Acary, Bernard Brogliato (“Implicit Euler numerical scheme and chattering-free implementation of sliding mode systems” in Systems and Control Letters, 2010) prove that the analytical and numerical solutions match after a finite number of samples at least for the case when there is no disturbance in control system. This result also extends to the multivariable case. These results relate to differential inclusions such as equation (8) in the cited ICML 2020 paper. Another recent relevant result on implicit numerical schemes is given by Brogliato et. al. (The Implicit Discretization of the Super-twisting Sliding-Mode Control Algorithm in IEEE Transactions on Automatic Control, 2020). As for the catch mentioned on explicit discretization, explicit Euler discretization results are given by Barbot et. al. (“Discrete differentiators based on sliding modes” in Automatica, 2020) for a discontinuous case arising in sliding modes. This reference establishes optimal accuracy asymptotics of their continuous-time counterparts. This reference also encompasses continuous non-Lipschitz right hand sides. In the presence of these results, we do not agree that it is superfluous to apply continuous finite-time methods to the case of DNN when several discretization methods are available, including the one proven in ICML 2020 reference.

---

### Author Response · Authors · 2020-11-24
**Manuscript and responses to reviews have been updated**

Dear Reviewers and all,

We have revised our manuscript based on all the comments provided to us. We thank the reviewers for their comments and would like to welcome them to review our updated manuscript and let us know their views/suggestions or any concerns.

With Regards,
Authors

---

### Decision · Program_Chairs · 2021-01-07
**Final Decision**

**Decision:**

Reject

**Comment:**

This paper aims to study the convergence of deep neural networks training via a control theoretic analysis. This is a very interesting approach to establish theoretical understanding of deep learning. However, there are several concerns raised by the reviewers:

1.	The contribution of this paper is limited. The results simply follow from standard optimal control. It is not clear what new insight the paper provides.
2.	There are already quite a few works on control theoretic analysis of deep learning. This paper did not do a good job on presenting its novelty and difference with existing works.
3.	The experimental part is weak. It only involves small data set and very simple networks.

Based on these, I am not able to recommend acceptance for the current manuscript. But the authors are encouraged to continue this research.